# Hidding the Ghostwriters: An Adversarial Evaluation of AI-Generated Student Essay Detection

**Xinlin Peng**[†1,2], **Ying Zhou**[†1,2], **Ben He**[*1,2], **Le Sun**[*2], **and Yingfei Sun**[*1]

[1]University of Chinese Academy of Sciences, Beijing, China
[2]Institute of Software, Chinese Academy of Sciences, Beijing, China
*pengxinlin22, zhouying20@mails.ucas.ac.cn,*
*benhe, yfsun@ucas.ac.cn, sunle@iscas.ac.cn*

## Abstract

Large language models (LLMs) have exhibited remarkable capabilities in text generation tasks. However, the utilization of these models carries inherent risks, including but not limited to plagiarism, the dissemination of fake news, and issues in educational exercises. Although several detectors have been proposed to address these concerns, their effectiveness against adversarial perturbations, specifically in the context of student essay writing, remains largely unexplored. This paper aims to bridge this gap by constructing AIG-ASAP, an AI-generated student essay dataset, employing a range of text perturbation methods that are expected to generate high-quality essays while evading detection. Through empirical experiments, we assess the performance of current AIGC detectors on the AIG-ASAP dataset. The results reveal that the existing detectors can be easily circumvented using straightforward automatic adversarial attacks. Specifically, we explore word substitution and sentence substitution perturbation methods that effectively evade detection while maintaining the quality of the generated essays. This highlights the urgent need for more accurate and robust methods to detect AI-generated student essays in the education domain. Code and data are released for public use[1].

## 1 Introduction

Large language models (LLMs) have demonstrated exceptional capabilities in various text generation tasks (Zhao et al., 2023; Wu et al., 2023; Li et al., 2023), raising both excitement and concerns within the research community and society at large. While these models offer tremendous potential for advancing natural language processing and understanding, their utilization also introduces inherent risks (Chen et al., 2023; Liu et al., 2023a; Sadasivan et al., 2023). Among these risks, there are particular challenges related to educational exercises.

For instance, students may rely on LLMs for essay writing practice, achieving high grades without developing essential writing skills. These risks highlight the importance of addressing the ethical and practical implications of LLM usage in order to mitigate potential negative consequences.

Efforts have been made to address these concerns by developing detectors for identifying AI-generated content (AIGC). The majority of existing studies in this area have focused on straightforward detection approaches (Liu et al., 2023b,a; Guo et al., 2023). There has been limited exploration of the effect of adversarial measures on detection accuracy (Susnjak, 2022; Liang et al., 2023), especially within the education domain. For example, Antoun et al. (2023) evaluate the robustness of detectors against character-level perturbations or misspelled words focusing on French as a case study. Krishna et al. (2023) train a generative model (DIPPER) to paraphrase paragraphs to evade detection. However, none of these studies are specific to the context of student essay writing in consideration of the quality of the generated text, as well as potential adversarial measures.

This paper aims to bridge this gap by presenting an adversarial evaluation of the effectiveness of AIGC detectors specifically on student essays. To facilitate this evaluation, we construct the AIG-ASAP dataset, a collection of AI-generated student essays based on the popular ASAP dataset[2] for automated essay scoring, using a variety of text perturbation methods. In addition to the existing paraphrasing perturbation method (Sadasivan et al., 2023), we address the observed tendency of LLMs to excessively use topical words in the generated essays by developing word substitution and sentence substitution methods that perturb the AI-generated essays at the word and sentence levels, respectively. The results demonstrate that while paraphrasing a human-written essay can help evade detection to

---

[†]Equal contribution.
[1]https://github.com/xinlinpeng/AIG-ASAP

[2]https://www.kaggle.com/c/asap-aes/

some extent, the word and sentence substitution perturbation methods significantly degrade the detection performance. Remarkably, these methods achieve this while preserving the high writing quality of the generated essays.

The major contributions of this research are threefold: 1) Proposal of the word substitution and sentence substitution perturbation methods that can evade AIGC detection while ensuring the quality of the generated essay. 2) We introduce the **AIG-ASAP** dataset, which serves as a benchmark for evaluating the performance of AIGC detectors in the education domain. By employing text perturbation techniques, we simulate real-world scenarios where ghostwriters or AI-powered tools are employed to produce essays unnoticeable by current detection methods. 3) We conduct empirical experiments to evaluate the performance of existing AIGC detectors on the AIG-ASAP dataset. Our findings shed light on the vulnerabilities of current detection methods, revealing that they can be easily circumvented through uncomplicated automatic adversarial attacks. This highlights the pressing need for more accurate and robust detection methods tailored to the unique challenges posed by AI-generated student essays.

## 2    Problem Statement

Artificial Intelligence Generated Content (AIGC) is defined as content that is generated by machines utilizing artificial intelligence (AI) systems, often in response to user-input prompts (Wu et al., 2023). In the upcoming sections, we conceptualize AIGC as a sequence-to-sequence generation process. Each input, denoted as $X = [I; C]$, comprises an instruction $I$ and contextual demonstrations $C$.

**Detection of AIGC.** With the recent advances of LLMs, AIGC has become increasingly proficient in mimicking human-generated content. Consequently, the detection of AIGC, which involves the task of differentiating between content generated by machines and content created by humans, has emerged as a significant area of research. Formally, the detection of AIGC can be defined as follows: Given a piece of textual content, represented as a sequence of tokens denoted by $X = [x_1, x_2, ..., x_n]$, where $x_i$ represents the $i$-th token, the task of AIGC detection aims to determine the probability that the content was generated by an AI system, denoted as $P(AIGC \mid X; \theta)$. The detection model parameters $\theta$ are trained to predict this probability, taking into account various linguistic and statistical features of the input text.

**AIGC Detection Attack.** AIGC detection attack refers to deceiving or evading AIGC detection through carefully crafted prompts (Lu et al., 2023) or paraphrasing attacks (Sadasivan et al., 2023), which aims to exploit vulnerabilities or limitations in the detection algorithms to evade detection or mislead the system into classifying AI-generated content as human-generated or vice versa. Given an AIGC detection model $P$ with parameters $\theta$, an attack on the AIGC detector aims to find an adversarial input $X^*$ that minimizes the detection probability of AIGC content $P(AIGC \mid X^*; \theta)$ while maximizing the probability of the content being classified as human-generated $P(Human \mid X^*; \theta)$.

## 3    Methodology

This section delves into the detailed process of constructing our AIG-ASAP dataset. Our study aims to investigate the vulnerability of AIGC detectors and their instability to perturbation for student essays, resulting in manipulated detection performance that decreased far from the original context.

### 3.1    Data Construction

Similar to the approach by Liu et al. (2023b), we create AIG-ASAP, a dataset of machine-generated essays, based on the existing ASAP dataset. ASAP is a well-established benchmark dataset utilized for evaluating automated essay scoring systems. It consists of essays written by high school students in the United States, covering eight different essay topics/prompts. Each essay in the dataset is accompanied by a human-assigned holistic score. We leverage the ASAP dataset and provide the LLM with the essay prompt to generate machine-written essays. The prompts serve as instructions or guidelines for the LLMs to generate essays in a controlled manner, as introduced in the following.

**Instruction-based Writing.** In recent research (Shyr et al., 2023; Qasem et al., 2023; Huang et al., 2023), it has been demonstrated that LLMs like ChatGPT exhibit strong language and logic abilities. To explore the potential of LLMs in content generation, we employ the original essay prompts in the ASAP dataset as input for the language model, considering the entirety of the LLM's output as the generated essay. In other words, we utilize the direct machine writing approach, where we solely provide the original essay topic require-

ments as the instruction $I$, without including any contextual information for $C$.

**Refined Writing.** In this data construction scenario, we provide the student-written essay as the reference for the LLMs to polish. By exposing the model to human-crafted essays, we strive to maintain control over factors such as length, format, and wording in the output of the LLM. This enables us to generate articles that are more in line with the writing style of students. We utilize the following sentence as instruction and the original full essay (referred to as $e_x$) as the context $C$.

> Polish and optimize the following essay: $e_x$.

**Continuation Writing.** Drawing inspiration from the pre-training paradigm of LLMs, we also adopt a left-to-right generation approach for the essay-generating task. To facilitate this, we offer the LLMs the initial first sentence of the ASAP essay (denoted as $e_{init}$) along with the composition requirements as the prompt, allowing the model to generate coherent and contextually appropriate essays based on the given input. Specifically, we consider the following prompt design:

> Follow the first couple of sentences from an essay to write a follow-up paragraph: $e_{init}$.

Examples of the prompts used in this work can be found in Appendix B.

### 3.2 Dataset Analysis

**The generated essays exhibit a decent level of diversity.** To assess the level of diversity in the generated essays, we conduct an analysis to measure the text overlap between pairs of essays. Specifically, we randomly select 100 essays for each topic and compute the SimHash similarity (Charikar, 2002) between each essay and all remaining essays for that topic. The histogram presented in Figure 1 provides insights into the distribution of SimHash similarities for essay pairs across the 8 topics, comparing machine-generated and human-written essays. The histogram shows that the similarity values of both types of essays roughly follow a normal distribution, spanning a range from 0.38 to 0.94. There is a moderate shift in the mode of the distributions, suggesting some differences in text overlap between the two types of essays. Furthermore, we calculate the average SimHash similarity of essay

pairs to be 0.677 in the AI-generated dataset and 0.617 in the human-written dataset. These values indicate a reasonable level of diversity among the AI-generated essays. Moreover, it is also observed that the machine-generated text after refining student essays exhibits higher similarity compared to a collection of human essays. This suggests that LLMs tend to utilize similar structures or expressions more frequently than humans.

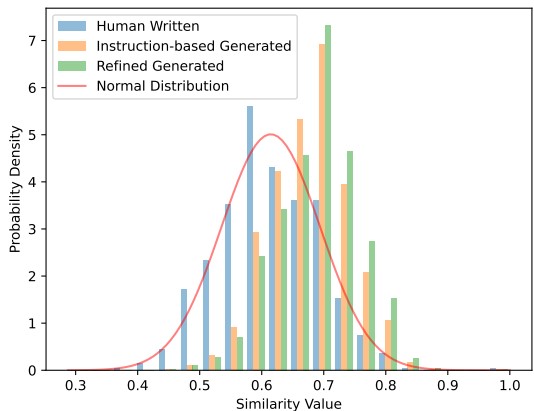

Figure 1: The SimHash probability distribution of human-written and ChatGPT-generated essays, with a normal distribution curve fitted to the data points of the human-written data.

**LLMs have a tendency to excessive use of topical words.** An empirical observation we made is that the LLMs often exhibit a tendency to repeatedly utilize words prompted in the essay topic while generating essays. For instance, when examining a set of 1,000 essays generated by ChatGPT using instruction-writing for ASAP topic 1, we find that the average frequency of occurrences of the topical word "technology" in each generated essay is 1.927, which is significantly higher than the average frequency of occurrences of 0.496 in each original human-written essay. This indicates that the LLMs tend to excessively employ topical words in their generated output. This excessive use of topical words opens up the possibility of evading detection by substituting these frequently used words with similar words or synonyms, without significantly degrading the quality of the generated essays. In the subsequent section, we will introduce the word and sentence substitution methods that explore this possibility and its impact on detection.

## 3.3 Perturbation Methods

Recent research (Krishna et al., 2023; Sadasivan et al., 2023) demonstrate that simple paraphrasing of AI-generated text, regardless of whether it originates from the same model or different models, can remarkably evade the detector. Therefore, to evaluate the stability of the current AI-generated text detector, we design and employ a range of text perturbation methods on our constructed AIG-ASAP dataset. Our study encompasses the following three methods: Essay Paraphrasing, Word Substitution, and Sentence Substitution. Within this set of methods, paraphrasing has been previously applied to the open-ended generation and long-form question answering. Empirical results indicate that the paraphrasing perturbation can be readily identified by a BM25 retrieve-then-compare detector (Krishna et al., 2023). However, its applicability to student essay writing is limited due to the requirement of a large set of machine-generated and human-written essays for comparison for each essay topic. In addition to paraphrasing, we design AI-generated text perturbation methods with two different granularities: word, and sentence. In essence, our perturbation method primarily concentrates on altering the distribution of the original generative model's output while preserving the quality of the essay, which aims to decrease the final detection performance to evade current detectors.

### 3.3.1 Essay Paraphrasing

According to Sadasivan et al. (2023), leveraging LLMs to rewrite the human-written text or pre-existing AI-generated text has the potential to disrupt the distribution of LLM outputs and the syntactic structure of the essay. Consequently, this can lead to a decreased probability of detection.

Previous research (Lu et al., 2023) has explored the use of LLMs to paraphrase content generated by other LLMs. However, our preliminary investigation suggests that this approach has minimal impact on both the writing quality and the detection rate of the generated essays. As a result, in this study, we shift our focus to rewriting genuine human-written essays instead, in order to more accurately replicate the writing process that students typically undergo. Essentially, to avoid detection, users often manually complete a low-quality draft composition with the assistance of the language model and then utilize the model to enhance the article. By incorporating these aspects into the AIG-

ASAP perturbation, the output text may exhibit greater coherence, mimicking the natural writing style of human authors and making it less distinguishable from genuine human-authored content. To guide the rewriting process of LLMs on human-written essays, given an essay $e_x$ from the ASAP dataset which is originally written by a human, we have designed prompt templates as follows:

> Please rewrite the essay and imitate its word using habits: $e_x$. Try to be different from the original text.

It is worth highlighting the intuitive similarity between refined writing and essay paraphrasing, as they both involve altering the distribution of the original output to affect the detection results. Nevertheless, our evaluation results reveal significant distinctions between the two approaches, which we will elaborate on in the experimental section.

### 3.3.2 Sentence Substitution

At the sentence level, we utilize a different and smaller generative model FLAN-T5 (Chung et al., 2022) to perform text replacement, in order to introduce diversity in the LLM-generated essays. For each LLM-generated essay, we randomly select a set of sentences and replace them with the mask token. Afterward, we employ the generative model to generate replacements for the masked segments. By incorporating random sentence replacements, we simulate a scenario where users manually modify certain parts of the AI-generated text with the assistance of a generative model. This enables us to assess the effectiveness of the generative model in reducing detection accuracy and emulating the writing process of human users. Formally, given an AI-generated essay $e_g$, a random masker $R$, and a smaller generative model $\mathcal{M}_{gen}$, we obtain the perturbed essay as follows:

$$\mathbf{e_1^*}, \mathbf{e_2^*}, ..., \mathbf{e_l^*} = \mathcal{M}_{gen}\left(R(\mathbf{e_g})\right) \quad (1)$$

### 3.3.3 Word Substitution

As aforementioned, we observe that LLMs tend to rely more on the information provided in the instructions compared to human writers when generating essays. Notably, the content generated by ChatGPT exhibits a significantly higher usage of words mentioned in the prompts compared to human-written essays. This discrepancy in writing behavior between LLMs and humans could

**Algorithm 1** Word Substitution

**Input**: Essay topic instruction $\mathcal{I}$, AI-generated essay $e$, knowledge base for synonyms $\mathcal{K}s$, smaller perturbation model $\mathcal{M}$

**Parameter**: $k$ number of words to be substituted, $p$ top predictions generated by $\mathcal{M}$, and $n$ maximal synonyms queried from $\mathcal{K}_h$

**Output**: Essay $e_{sub}$ with at most $k$ substituted words

1: **Stage 1: High-frequency Words Sampling**
2: Initialize $\mathcal{W}_h$ to store high-frequency words: $\mathcal{W}_h \leftarrow \{\}$
3: **while** $|\mathcal{W}_h| < k$ **do**
4:    $w \leftarrow$ Sample a high-frequency word from the instruction $\mathcal{I}$ or the essay $e$.
5:    **if** $w$ not in $\mathcal{W}_h$ **then**
6:       Add $w$ to $\mathcal{W}_h$
7:    **end if**
8: **end while**
9: **Stage 2: Word Substitution with $\mathcal{K}_s$ and $\mathcal{M}$**
10: **for** $w$ in $\mathcal{W}_h$ **do**
11:    $w_{\mathcal{M}}^{sub} \leftarrow$ Generate and sample top $p$ candidates by replacing all occurrences of $w$ in $e$ with the [MASK] token using the perturbation model $\mathcal{M}$
12:    $w_{\mathcal{K}_s}^{sub} \leftarrow$ Query the knowledge base $\mathcal{K}s$ to retrieve up to $n$ synonyms for $w$
13: **end for**
14: Replace each $w$ in $\mathcal{W}_h$ with corresponding words in $w_{\mathcal{M}}^{sub} \cap w_{\mathcal{K}s}^{sub}$ to generate $e_{sub}$
15: **return** $e_{sub}$

---

potentially account for this observed phenomenon. Based on these insights and drawing from the research of robustness in natural language processing (Antoun et al., 2023), we devise a perturbation strategy that substitutes the frequent words in the prompts. An overview of our word substitution procedure is outlined in Algorithm 1. The algorithm starts by identifying the high-frequency words in each topic's prompt. These words are then replaced with the [MASK] token, and the BERT-base model (Devlin et al., 2019) is utilized to predict the most suitable substitution for each masked word. To ensure the diversity and quality of the generated essays while maintaining semantic coherence and relevance to the given topic, we leverage Word-Net [3] to find similar words as candidates for each masked word. If WordNet does not provide any

---

[3] https://wordnet.princeton.edu/

candidate, we select the top-1 word based on the BERT prediction score.

We have introduced three methods for perturbing essays, each contributing to a perturbation approach that ranges from coarse-grained to fine-grained techniques. The first method we employ operates at a coarse-grained level, involving full-text replacement through paraphrasing. By rewriting the entire content, we are able to introduce substantial changes to the original essay while maintaining its overall structure and coherence. Moreover, we take a finer-grained approach by focusing on the manipulation of individual sentences. By selecting and modifying specific sentences, we can introduce subtle changes to the article's content and meaning. Lastly, based on our observations of the distinct writing styles between humans and LLMs, we introduce to operate at a more granular level, targeting word substitution. By analyzing high-frequency words and utilizing a perturbation model, we selectively replace specific words in the essay with suitable alternatives. This approach enables more nuanced perturbations while preserving the original structure and context of the essay.

## 4 Experiments

In this section, we evaluate the performance of detectors on AIG-ASAP to assess their accuracy and AUROC in detecting AI-generated essays.

### 4.1 Experimental Setup

**Datasets.** Based on ASAP, we have created several sets of AI-generated essays: *AIG-ASAP-instruction-based-writing*, *AIG-ASAP-refined*, and *AIG-ASAP-continuation*, based on the different generation stages as stated in Section 3.1. Additionally, by introducing perturbations to the *AIG-ASAP-instruction-based-writing* dataset using BERT-base (Devlin et al., 2019) for word substitution and FLAN-T5-base (Chung et al., 2022) for sentence substitution, we generate three sets of challenging AIG-essays: *AIG-ASAP-paraphrasing*, *AIG-ASAP-word-substitution*, and *AIG-ASAP-sentence-substitution*. For word substitution, we choose the top-10 frequent words to be replaced. For sentence substitution, we randomly select 20% of the sentences within each essay to be masked and completed using FLAN-T5-base. For brevity, we will omit the prefix *AIG-ASAP* from the dataset names in the subsequent sections. For training purposes, we use a mixture of the randomly selected 90%

of the generated essays for each of the machine-generated essay sets. The remaining 10% of the generated essays are used for testing. As for the human-written essays, we also randomly select 90% for training, and use the rest for testing. Statistics of the dataset can be found in Appendix A.

**Generators.** In order to better simulate the usage scenarios of student users in educational applications, we employ several commonly used open-source or commercial LLMs for essay generation and perturbation, including ChatGPT[4], GPT-4[5], and Vicuna-7b (Chiang et al., 2023).

**Detectors.** We evaluate the performance of open-source detectors over our dataset, such as ArguGPT (Liu et al., 2023a) and CheckGPT (Liu et al., 2023b). In particular, the recent RoBERTa-QA and RoBERTa-Single models trained on the HC3 dataset (Guo et al., 2023) serve as state-of-the-art detectors as they show strong detection performance in previous studies (Liu et al., 2023a; Wang et al., 2023b). Furthermore, we can gain further insights into the performance variations by fine-tuning the RoBERTa-QA/Single detectors using the training set of our constructed data.

**Essay Scorer.** We incorporate text quality of the generated essays in our evaluation. In reference to recent work on AES (Jin et al., 2018; Wang et al., 2022), we train a straightforward yet highly effective scorer. We initially normalize the score range of various essay topics to values between 0 and 10. Subsequently, we employ RoBERTa-base (Liu et al., 2019) to train a scorer on the original ASAP training data. On the reserved validation data, the fine-tuned scorer achieves a quadratic weighted kappa (QWK) of 0.770, which is close to the average inter-human QWK agreement of 0.760 reported in (Doewes and Pechenizkiy, 2021).

**Human Evaluation.** We further conduct human evaluation tests by presenting two annotators (master candidates in computer science) with 162 sets of essay pairs (an average of 9 essay pairs are evaluated in each subset of AIG-ASAP) and recording their pairwise preferences, without disclosing which essays were written by humans or AI. Then statistical analyses are performed to assess inter-annotator agreement and draw conclusions about the quality of essays generated by humans and AI.

**Metrics.** In line with previous studies (Antoun et al., 2023; Chakraborty et al., 2023; Liu et al.,

2023a,b), we report the detection accuracy of both the AI-generated and the human-written essays to evaluate the performance of the detector in distinguishing between genuine human-written content and AI-generated content. In consideration of evaluating the confidence of the detectors, we also offer the area under the receiver operating characteristic (AUROC) (Fawcett, 2006) results. Furthermore, we incorporate the AES score as an additional criterion to evaluate the quality of the generated essays.

## 4.2 Results

**AI-generated essays without perturbation are highly identifiable.** The detection accuracy and AUROC of essays generated by different language models under various detectors are summarized in Table 1. As shown in Table 1, the main conclusion drawn is that machine-generated essays without perturbation can be easily detected by all detectors, regardless of whether they are written directly based on instructions, refined by human articles, or continued from an existing text. As for the best-performing detector, RoBERTa-QA, it achieves detection accuracy and AUROC of over 90% for nearly all machine-generated data. Notably, it also demonstrates a decent accuracy of 89.3% when detecting human-written essays.

**Rewriting and substitution greatly reduce detection accuracy.** As expected, the application of perturbed methods to machine-generated text effectively reduces the possibility of detection. We observe that as the granularity of perturbation decreases from full-text to word, the impact on the detector's performance becomes more evident. Remarkably, using the word level perturbation, in many cases, the detection accuracy is reduced to approximately 50%, which is almost equivalent to a random classifier. Furthermore, as discussed in Section 3.3.1, both refined and paraphrasing perturbations require a given human-written essay to generate. However, in terms of detection effectiveness, *paraphrasing* demonstrates better attack performance. We speculate that this performance difference comes from the prompt design. In paraphrasing, the given article serves more as a reference demonstration rather than a strict template. Consequently, the language model has the flexibility to incorporate more diversity and creativity in its writing, leading to improved evasion of detectors.

**Finetuned RoBERTa detectors can identify rewritten essays but fail to detect essays that**

[4]https://chat.openai.com
[5]https://openai.com/gpt-4

| Generator/+Perturb. | ArguGPT 0-shot | | CheckGPT 0-shot | | RoBERTa-Single 0-shot | | fine-tune | | RoBERTa-QA 0-shot | | fine-tune | | Quality score |
| | ACC | AUC | ACC | AUC | ACC | AUC | ACC | AUC | ACC | AUC | ACC | AUC | (0-10) |
|---|---|---|---|---|---|---|---|---|---|---|---|---|---|
| Human | 69.2 | 68.8 | 99.5 | 90.7 | 90.9 | 88.4 | 90.5 | 87.2 | 89.8 | 85.3 | 89.3 | 86.8 | 6.25 |
| Vicuna-Instr-writing | 93.5 | 92.3 | 92.8 | 92.4 | 76.2 | 74.3 | 78.0 | 75.7 | 94.2 | 91.2 | 94.8 | 91.9 | 6.08 |
| -refined | 88.3 | 87.2 | 88.5 | 87.3 | 72.1 | 70.1 | 76.5 | 74.3 | 91.9 | 89.6 | 90.3 | 87.4 | 5.81 |
| -continuation | 90.7 | 88.5 | 93.4 | 89.6 | 69.7 | 67.4 | 73.1 | 71.5 | 92.5 | 90.8 | 92.5 | 91.5 | 5.90 |
| +paraphrasing | 91.0 | 90.0 | 89.9 | 87.9 | 74.3 | 73.8 | 77.2 | 76.9 | 89.1 | 88.2 | 88.6 | 88.6 | 5.79 |
| +sentence-sub | 74.7 | 72.6 | 60.9 | 60.5 | 59.2 | 58.5 | 62.0 | 63.3 | 63.6 | 64.6 | 66.8 | 68.5 | 6.12 |
| +word-sub | **62.5** | **60.8** | **57.2** | **56.9** | **47.4** | **46.8** | **52.6** | **54.3** | **51.2** | **53.7** | **53.9** | **57.2** | 6.02 |
| ChatGPT-Instr-writing | 95.4 | 94.8 | 90.6 | 89.3 | 69.6 | 68.7 | 81.5 | 82.4 | 93.3 | 93.8 | 94.1 | 94.8 | 8.20 |
| -refined | 94.6 | 93.5 | 89.6 | 88.9 | 64.3 | 63.0 | 80.9 | 82.7 | 90.6 | 90.9 | 92.0 | 92.4 | 8.31 |
| -continuation | 97.7 | 96.1 | 89.1 | 87.3 | 79.1 | 76.2 | 82.8 | 81.1 | 95.5 | 94.6 | 95.9 | 96.1 | 7.27 |
| +paraphrasing | 84.6 | 82.8 | 72.1 | 69.7 | 53.1 | 52.8 | 74.6 | 73.3 | 71.8 | 73.9 | 82.7 | 84.0 | 8.21 |
| +sentence-sub | 61.0 | 60.3 | 57.3 | 57.6 | 62.9 | 63.5 | 65.8 | 69.2 | 64.7 | 68.8 | 68.1 | 72.3 | 7.36 |
| +word-sub | **53.8** | **53.4** | **49.9** | **51.3** | **55.0** | **53.6** | **58.2** | **60.3** | **57.4** | **60.8** | **61.6** | **64.7** | 8.12 |
| GPT-4-Instr-writing | 96.9 | 94.8 | 92.4 | 91.5 | 71.3 | 70.4 | 80.6 | 82.5 | 93.1 | 93.0 | 93.7 | 94.2 | 8.46 |
| -refined | 92.0 | 90.1 | 85.3 | 86.0 | 68.2 | 67.7 | 72.2 | 74.7 | 89.1 | 89.6 | 88.7 | 89.2 | 8.63 |
| -continuation | 95.2 | 93.7 | 88.9 | 87.5 | 73.0 | 72.1 | 78.1 | 78.7 | 91.7 | 92.4 | 90.2 | 95.1 | 8.35 |
| +paraphrasing | 82.3 | 80.5 | 71.8 | 72.6 | 66.4 | 67.1 | 75.3 | 76.3 | 69.7 | 70.5 | 78.5 | 79.7 | 8.17 |
| +sentence-sub | 64.9 | 62.7 | 58.5 | 59.2 | 61.3 | 63.1 | 63.3 | 64.7 | 58.0 | 59.4 | 63.8 | 65.3 | 7.82 |
| +word-sub | **55.6** | **54.3** | **53.6** | **53.8** | **54.9** | **55.7** | **58.1** | **59.4** | **56.2** | **57.3** | **59.5** | **61.7** | 8.21 |

Table 1: Accuracy(ACC) and AUROC(AUC) of various detectors using different essay generation strategies. As the perturbation granularity becomes finer, there is a decrease in the probability of AI-generated text being detected. "0-shot" represents the performance of the off-the-shelf detector, "fine-tune" corresponds to our model further fine-tuned with training data, and "quality" of the generated essays is measured using our trained essay scorer. The lowest detection rate to a generator of each detector is in **bold**. The lowest quality AES score of each generator is underlined.

| Metric | Evaluator | Instr-writing | Refined | Cont. | Paraph. | Sent-sub | Word-sub |
|---|---|---|---|---|---|---|---|
| AIGC | $scorer_{aes}$ | 70.4 | 77.8 | 63.0 | 74.1 | 66.7 | 63.0 |
| Pref. | $human_1$ | 63.0 | 74.1 | 59.3 | 66.7 | 55.6 | 44.4 |
| Ratio | $human_2$ | 66.7 | 70.4 | 59.3 | 74.1 | 63.0 | 55.5 |
| Evaluators' Kappa | $human_1$ & $scorer_{aes}$ | 66.9 | 70.0 | 76.7 | 82.4 | 46.2 | 35.2 |
| | $human_2$ & $scorer_{aes}$ | 74.3 | 80.9 | 76.7 | 61.4 | 59.5 | 54.2 |
| | $human_1$ & $human_2$ | 91.9 | 72.4 | 84.7 | 82.4 | 84.7 | 78.0 |

Table 2: Results of human evaluation. AIGC Pref. Ratio refers to the ratio of AI-generated text being preferred by the evaluator. The kappa value that measures the inter-rater agreement is also given.

**underwent substitution.** To better identify the perturbed essays, we proceed to further fine-tune the RoBERTa-Single and RoBERTa-QA using the reserved training data, as the method described in the original paper. As shown in the "fine-tune" columns of Table 1, the detectors have successfully improved their ability to detect machine-generated content while maintaining their proficiency in identifying human-written essays. Notably, for the ChatGPT paraphrasing data, RoBERTa-Single achieves a notable improvement, increasing from 54.1% to 74.6% in terms of detection accuracy. However, it should be noted that the performance improvement achieved through fine-tuning varies across the perturbation methods. In the case of

the substitution methods, the fine-tuned model has resulted in a relatively modest improvement, with an increase of only up to 5.2% in the performance which is evident from the Vicuna word substitution data on the RoBERTa-Single model. In particular, the accuracy of the fine-tuned RoBERTa models is still around or below 60% of the generated essays that underwent word substitutions, which signals an unreliable detection performance for practical use.

**Human quality evaluation.** Table 2 presents the results of the human evaluation conducted to assess the quality of the generated essays. The results from the AIGC Pref. Ratio show that, in general, AI-generated essays are more likely to be preferred

| Essay Type | Instr-writing | Refined | Cont. | Paraph. | Sent-sub | Word-sub |
|---|---|---|---|---|---|---|
| Argumentative | 92.7 | 91.3 | 93.8 | 80.7 | 63.4 | **38.4** |
| Source-dependent | 95.1 | 94.5 | 97.6 | 84.1 | 72.6 | 74.7 |
| Narrative | 94.5 | 90.2 | 96.3 | 83.2 | 68.3 | 58.6 |

Table 3: The detection accuracy of RoBERTa-QA (fine-tuned) varies across different essay types. All perturbation methods demonstrate moderate fluctuations across various essay categories, while word substitution has a notable impact on argumentative essays.

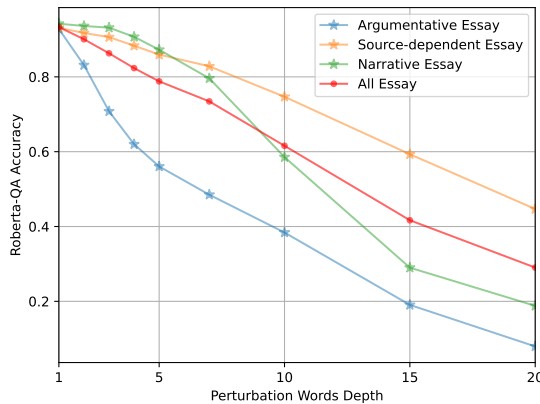

Figure 2: Detection performance curves for different essay categories using RoBERTa-QA (fine-tuned), where the X-axis represents perturbation depth.

by both human and machine evaluators, particularly for essays generated through refined writing and paraphrasing techniques. However, when it comes to word substitution, only a half chance of preference is observed. This can be attributed to the complete replacement of frequent words in the essays, which may be inappropriate in certain cases. Although the word substitution experiments demonstrate the potential for circumventing AIGC detectors, the human evaluation results suggest that further improvements can be made to enhance the quality of the generated essays. For instance, a partial replacement of frequent words could be considered to preserve the writing quality of language models while introducing perturbations. Furthermore, the results indicate a substantial degree of consistency between human evaluation and the automatic scorer. However, on the word or sentence substitution datasets, the consistency is limited, suggesting a certain level of discrepancy between the viewpoints of humans and algorithms.

### 4.3 Analysis

**Detection difficulty varies to different essay types.** The AIG-ASAP dataset encompasses 8 distinct topics that are further categorized into 3 essay types: argumentative, source-dependent, and narrative. Detecting these essay topics reveals variations in the difficulty of AIGC detection and the impact of perturbations. Table 3 shows the detection accuracy across essay categories, focusing on the best-performing detector, fine-tuned RoBERTa-QA, when applied to ChatGPT-generated data. The results indicate that, on average, source-dependent essays are more easily detectable by the detector. This observation could be attributed to the source article that is provided to ChatGPT during source-dependent content generation. Such source articles may impose limitations on the diversity of words used by the language model, making the generated content more distinguishable. Additionally, it is observed that the detection performance over essay paraphrasing and sentence substitution has moderate fluctuations over different essay categories. However, word substitution has a significant influence on argumentative essays, which can be attributed to the fact that the words selected for substitution in argumentative essays are more likely to involve argumentative keywords, resulting in a more noticeable effect on the overall content.

**Perturbation Depth.** We further investigate the influence of increasing perturbation depth on the detection accuracy of AI-generated essays with word substitutions, where perturbation depth refers to the number of most frequent words replaced with perturbation methods while generating *AIG-ASAP-word-substitution* set. Figure 2 presents the relationship between perturbation depth and detection accuracy. The curves clearly demonstrate an inverse relationship between perturbation depth and detection probability, confirming the intuitive expectation that the more perturbed words present, the lower the likelihood of detection by the detector. Additionally, it is worth noting that the quality of the perturbed content experiences a slight decrease, as detailed in Appendix D.

## 5 Related Work

With the advances of large language models (LLMs), there have been a plethora of research on detecting AI-generated content (Becker et al., 2023; Crothers et al., 2022; Jawahar et al., 2020; Mireshghallah et al., 2023; Su et al., 2023; Wang et al., 2023a). Current AI-generated text detection methods can be in general categorized into three categories (Lu et al., 2023): statistical (Gehrmann et al., 2019), watermarking (Kirchenbauer et al., 2023), and training-based (Gallé et al., 2021).

The statistical approach analyzes measures like entropy, perplexity, and $n$-gram frequency to identify statistical irregularities (Gehrmann et al., 2019; Lu et al., 2023), which rely on statistical characteristics of the text to differentiate between AI-generated and human-written content. The watermarking methods (Wilson et al., 2014) imprint specific patterns on generated text. Soft watermarking (Kirchenbauer et al., 2023) presents a novel statistical test designed to detect the presence of a watermark, providing interpretable p-values as evidence, and additionally, establishing an information-theoretic framework that enables the analysis of the watermark's sensitivity. Early training-based detection studies focused on identifying fake reviews (Hovy, 2016) and fake news (Zellers et al., 2019). Recent research employed LLM-generated texts for training classifiers, such as ArguGPT (Liu et al., 2023a), CheckGPT (Liu et al., 2023b), and G3Detector (Zhan et al., 2023). Notably, Guo et al. (2023) fine-tuned the RoBERTa-QA and RoBERTa-single models on the HC3 dataset and achieved high detection accuracy.

## 6 Conclusion

We note that basic paraphrasing techniques like rewriting essays can decrease the effectiveness of detection, while fully automated machine-generated content can be easily detected by current detectors. This implies that human writing plays a crucial role in evading detection in the traditional process. To address this, we introduced two distinct perturbation strategies: word substitution and sentence substitution. These strategies have shown substantial potential in deceiving AI-generated content detection while ensuring content quality in our experiments. The results highlight the inherent vulnerabilities in existing detection methods and emphasize the need for the development of more resilient and precise approaches for detecting AI-generated student essays.

## Acknowledgments

This work is supported by the National Natural Science Foundation of China (62272439), and the Fundamental Research Funds for the Central Universities.

## Limitations

This paper addresses an important and timely topic regarding the detection of AI-generated student essays. However, the AIG-ASAP dataset constructed for this research is built upon the ASAP dataset composed of English essays written by high-school students in the United States. The dataset's effectiveness in representing the broader landscape of AI-generated essays in different educational contexts and different languages could impact the generalizability of the findings.

While the paper emphasizes the need for more accurate and robust methods to detect AI-generated student essays, it does not delve into the practical implementation and deployment of such methods. Considering the potential impact on educational institutions and the challenges of real-time detection, further exploration of the feasibility and scalability of proposed detection methods would enhance the practical implications of the research.

## Ethics Statement

In this paper, we investigate the potential vulnerability concerns of the current AIGC detection methods and explore several adversarial measures that generate student essays that are not easily detected. We hope that this study could inspire further exploration and design of AIGC detection methods and aid in the development of robust real-world AIGC detectors.

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

## A   Dataset Statistics

Statistics of our constructed AIG-ASAP dataset can be found in Table 4.

## B   AIG-ASAP Data Construction Prompts

### B.1   Instruction-base Writing

**Argumentative prompt - Topic 1**   Act as a middle school student, please read the below prompt and write an essay:

More and more people use computers, but not everyone agrees that this benefits society. Those who support advances in technology believe that computers have a positive effect on people. They teach hand-eye coordination, give people the ability to learn about faraway places and people, and even allow people to talk online with other people. Others have different ideas. Some experts are concerned that people are spending too much time on their computers and less time exercising, enjoying nature, and interacting with family and friends.

Write a letter to your local newspaper in which you state your opinion on the effects computers have on people. Persuade the readers to agree with you.

**Source-dependent prompt - Topic 3**   Act as a middle school student, please read the source essay and write an essay based on given topic prompt.

Source essay:

"ROUGH ROAD AHEAD: Do Not Exceed Posted Speed Limit by Joe Kurmaskie FORGET THAT OLD SAYING ABOUT NEVER taking candy from strangers. No, a better piece of advice for the solo cyclist would be, "Never accept travel advice from a collection of old-timers who haven't left the confines of their porches since Carter was in office." It's not that a group of old guys doesn't know the terrain. With age comes wisdom and all that, but the world is a fluid place. Things change..."

Prompt:

Write a response that explains how the features of the setting affect the cyclist. In your response, include examples from the essay that support your conclusion.

**Narrative prompt - Topic 7**   Act as a middle school student, please read the below prompt and write an essay:

Write about patience. Being patient means that you are understanding and tolerant. A patient person experience difficulties without complaining. Do only one of the following: write a story about a time when you were patient OR write a story about a time when someone you know was patient OR write a story in your own way about patience.

### B.2   Refined Writing

**General Prompt**   Polish and optimize the following essay:

Based on the story it all started when they had build the empire state building and the workers should have knowned that the building was too big for the wind climate, up in the sky and plus when you make a building like that you have to make sure it's strong enough to keep balance. They faced problem's like when the U.S. tried to make a deal with them if they would let them use the top of the building for a dock to let civilians on the deck, and after they made a law that no more blimps can land on the dock a blimp attempted to land on the dock.

### B.3   Continuation Writing

**General prompt**   Follow the first couple of sentences from an essay to write a follow-up paragraph:

Dear @LOCATION1, I know having computers has a positive effect on people....

Besides, consider the essay writing prompt:

More and more people use computers, but not everyone agrees that this benefits society. Those who support advances in technology believe that computers have a positive effect on people. They teach hand-eye coordination, give people the ability to learn about faraway places and people, and even allow people to talk online with other people. Others have different ideas. Some experts are concerned that people are spending too much time on their computers and less time exercising, enjoying nature, and interacting with family and friends.

Write a letter to your local newspaper in which you state your opinion on the effects computers have on people. Persuade the readers to agree with you.

| Perturbation | | Human | Vicuna | ChatGPT | GPT-4 |
|---|---|---|---|---|---|
| Instr-writing | train | 5114 | 3600 | 7200 | 3600 |
| | test | 571 | 400 | 800 | 400 |
| Refined | train | - | 5114 | 10230 | 5114 |
| | test | - | 571 | 1140 | 571 |
| Cont. | train | - | 5114 | 10230 | 5114 |
| | test | - | 571 | 1140 | 571 |
| Paraph. | train | - | 5114 | 5114 | 5114 |
| | test | - | 571 | 571 | 571 |
| Sent-sub | train | - | 3600 | 7200 | 3600 |
| | test | - | 400 | 800 | 400 |
| Word-sub | train | - | 3600 | 7200 | 3600 |
| | test | - | 400 | 800 | 400 |

Table 4: Statistics of the AIG-ASAP dataset.

## C  Substitution examples

We give an example of the result essays under word substitution and sentence substitution. The essay after word substitution is detected as 'Human' with the 0.67 probability and the essay after sentence substitution is detected as 'Human' with the 0.81 probability while the instruction-based writing result is detected as 'ChatGPT' with the 0.99 probability. The quality score received from our essay scorer is 8.33 for instruction-based writing, 8.18 for word substitution, and 8.05 for sentence substitution. The substitutions may not be the most appropriate alternatives as shown in the examples, leading to degraded quality as noticed by the essay scorer, but drastically reduces the chance of being caught by the detector.

**Instruction-based writing result**  Patience is something that I have always struggled with. I am someone who always wants things to happen quickly and tend to get frustrated when they don't. However, witnessing the patience of someone else has helped me to understand the importance of being patient.

Last summer, my grandpa suffered a stroke and was admitted to the hospital. It was a difficult time for my family as we watched him struggle to do things that were once easy for him. However, my grandma was the epitome of patience. She never once complained or got frustrated with my grandpa, no matter how difficult things became.

I remember one day when we were visiting him in the hospital, my grandpa was having a hard time speaking, and it was taking him a long time to get his words out. My grandma sat beside him, listening intently and waiting patiently for him to finish his sentences. She didn't interrupt him or finish his thoughts for him; she simply waited until he was done speaking.

Her patience not only helped my grandpa feel heard and understood, but it also helped our entire family to stay calm and positive during a difficult time. Whenever I feel myself getting impatient, I think back to that moment in the hospital room and try to channel my grandma's patience.

Being patient may not always be easy, but it is an incredibly valuable trait to have. It allows us to be more understanding and tolerant of others and helps us to experience difficulties without complaining.

**Word substitution result**  Patience is something that I have always struggled with. I am somebody who always wants things to happen quickly and tend to gain frustrated when they don't. However, witnessing the tolerance of somebody else has take me to understand the importance of being patient.

Last summer, my grandpa suffered a stroke and was admitted to the hospital. It was a difficult task for my family as we watched him struggle to do things that were once easy for him. However, my grandma was the epitome of taught. She never once complained or got frustrated with my grandpa, no matter how difficult things are.

I remember a night when we were visiting him in the hospital, my grandpa was having a hard strongless speaking, and it was taking him a long effect

to gain his words out. My grandma sat beside him, listening intently and awaiting patiently for him to finish his sentences. She didn't interrupt him or complete his thoughts for him; she simply timeed until he was dtime speaking.

Her tolerance not only make my grandpa feel heard and understood, but it also make our entire family to stay calm and effective during a difficult time. Whenever I feel myself become impatience, I think back to that moment in the hospital room and try to channel my grandma's patience.

Being patience may not always be easy, but it is an incredibly valuable trait to have. It allows us to be more understanding and tolerant of others and make us to experience difficulties without complaining.

**Sentence substitution result**   Patience is something that I have always struggled with. I am someone who always wants things to happen quickly and tend to get frustrated when they don't. And my grandma was the epitome of patience.

Last summer, my grandpa suffered a stroke and was admitted to the hospital. As my grandma's patience has helped me understand the importance of being patient. However, my grandma was the epitome of patience. She never once complained or got frustrated with my grandpa, no matter how difficult things became.

I'm always frustrated when things don't happen quickly. I'm always frustrated when things don't go as planned. I remember one day when we were visiting him in the hospital, my grandpa was having a hard time speaking, and it was taking him a long time to get his words out. My grandma sat beside him, listening intently and waiting patiently for him to finish his sentences. My grandma was a patient and understanding person. Whenever I feel myself getting impatient, I think back to that moment in the hospital room and try to channel my grandma's patience.

Being patient may not always be easy, but it is an incredibly valuable trait to have. It allows us to be more understanding and tolerant of others and helps us to experience difficulties without complaining.

## D   Detailed Analysis

**Context Quality with Perturbations.**   According to the previous examples, the substitutions may introduce spelling or grammatical errors, leading to degraded quality as noticed by the essay scorer, but drastically reduce the chance of being caught

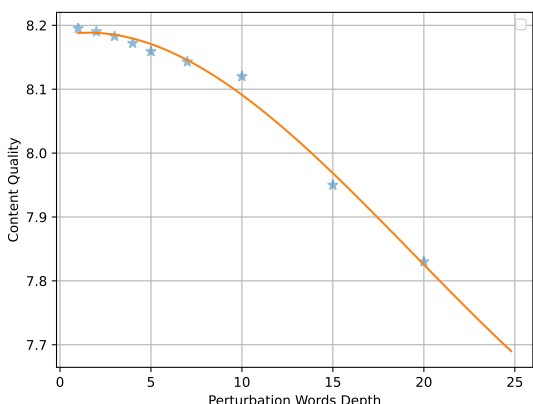

Figure 3: Our proposed perturbation methods result in a slight decrease in content quality. Even after replacing 20 words, the overall essay scores only drop by 0.365.

by the detector. To explore the extent of this quality degradation, we present the corresponding curve in Figure 3. It can be observed that due to the comprehensive consideration of synonyms and language model output predictions during perturbation, the score of the perturbed text does not decrease rapidly as the perturbation depth increases. This indicates that the quality of the perturbed text is relatively preserved, despite the increasing level of perturbation applied.