# OpenReview forum: "Hidding the Ghostwriters: An Adversarial Evaluation of AI-Generated Student Essay Detection"
_EMNLP/2023/Conference — EMNLP 2023 Main_

### Official Review · Reviewer_WQAB · 2023-08-03

**Typos Grammar Style And Presentation Improvements:** Line 418 purposes, we...
**Soundness:** 4

**Excitement:**

4: Strong: This paper deepens the understanding of some phenomenon or lowers the barriers to an existing research direction.

**Paper Topic And Main Contributions:**

This paper constructs a new data resource AIG-ASAP, which employs a range of text perturbation methods and can be used for evaluating AI-generated content detectors. It also assesses the performance of current AIGC detectors on this dataset through empirical experiments.

**Questions For The Authors:**

In the results, you list the accuracy of binary classification, right? But do the detectors always provide only binary results(i.e. human written or Ai-generated)? Have you considered evaluating the confidence of the detectors? For example, the probability of one essay being AI-generated is  xx%. With such confidence-related metrics, it would be interesting to see how much the performance was dropped through adversarial attacks.

**Reasons To Accept:**

Solid work with details of the generation process. Not only the dataset could be reused, but also the methods are highly reproducible. Important findings highlight the vulnerabilities in existing detection methods.

**Reasons To Reject:**

No discussion on how to improve the robustness of detectors to this dataset. The quality of the AI-generated essays was automatically scored.

**Reproducibility:**

5: Could easily reproduce the results.

**Reviewer Confidence:**

4: Quite sure. I tried to check the important points carefully. It's unlikely, though conceivable, that I missed something that should affect my ratings.

---

> ### Author Rebuttal · Authors · 2023-08-29
>
> Thanks for your constructive comments. Please see below for our responses.
>
> 1. Robustness. While our paper primarily focuses on the adversarial evaluation of AIGC detectors, we recognize the significance of addressing practical implementation and robustness. We believe that utilizing reinforcement learning and adversarial generation techniques to address perturbation challenges are viable future research directions.
> 2. Quality evaluation. Given the substantial number of AI-generated essays, we employed an automatic scorer for quality evaluation, which demonstrated a high kappa agreement (0.77) with human annotators. In a future version of our work, we plan to add human-evaluation results such as pairwise preferences. This involves presenting annotators with sets of essay pairs, without disclosing which essays were written by humans or AI.
> 3. Threshold. Following [1], we selected the default confidence threshold setting (0.5) for our detectors for binary classification. We plan to include additional experiments in a revised version to evaluate other confidence-related metrics.
>
> [1] Biyang Guo, Xin Zhang, Ziyuan Wang, Minqi Jiang, Jinran Nie, Yuxuan Ding, Jianwei Yue, and Yupeng Wu. 2023. "How close is chatgpt to human experts? comparison corpus, evaluation, and detection "

---

### Official Review · Reviewer_rqmL · 2023-08-03

**Soundness:** 2

**Excitement:**

4: Strong: This paper deepens the understanding of some phenomenon or lowers the barriers to an existing research direction.

**Paper Topic And Main Contributions:**

The paper introduces an adversarial dataset to evade model detection accuracy in student essay writing domain. They use basic adversarial attacks such as word and sentence perturbation techniques on GPT models while preserving the quality. Their analyses show that models generally fail in the detection performance when using these attacks.

**Reasons To Accept:**

This paper introduces a novel application of adversarial attacks in the student essay domain that opens a new research direction across the education and AI field. It also attempts to resolve the concerns on ethical and practical usage of LLMs in other domains. Also, the paper is logically sound and includes comprehensive analysis to support its argument.

**Reasons To Reject:**

High quality.
The authors state that the generated essays using three adversarial techniques maintain the quality of the original essays. Were there any qualitative results that confirm the quality of the original and generated essays are comparable?  Also, GPT-instructed/created sentences are known to have textual --grammatical, syntactical--noise (which is obvious to humans). Was there any process to filter these out?

Selection of Topical Words.
The authors state that topical words were replaced in the generated essays to create new essays. The paper does not mention how these topical words were selected and replaced. To my understanding, only the overlapping words with the prompts were replaced. How does removing topical words affect the detection accuracy?

Quality Metric.
It seems like the metric from asap-aes was only employed to measure the quality of the essays. It might be worthwhile to adopt other metrics for essay evaluation, as adversarial samples created from LLMs could be biased in a way that could lead to lexical hallucinations, and lower the detection accuracy.

Dataset Analysis.
The authors do not provide elementary distribution of the introduced dataset, such as topics, types, sentence length of essays, and number of words or sentences perturbed in each essay. Dataset papers should include basic information about the dataset.

The wording could be improved to explain the concept better and repetitive information throughout the paper could be removed.


**Reproducibility:**

4: Could mostly reproduce the results, but there may be some variation because of sample variance or minor variations in their interpretation of the protocol or method.

**Reviewer Confidence:**

3: Pretty sure, but there's a chance I missed something. Although I have a good feel for this area in general, I did not carefully check the paper's details, e.g., the math, experimental design, or novelty.

**Typos Grammar Style And Presentation Improvements:**

No typos or grammar mistakes, but the organization of the paper could be improved. It was a bit confusing to understand the task intuitively. The paragraphs could be more self-contained.

---

> ### Author Rebuttal · Authors · 2023-08-29
>
> Thanks for your constructive comments. Please see below for our responses.
>
> 1. High quality. We used automatic quality evaluation due to the large number of essays. Essays in ASAP were written by high-school students which have highly diverse human ratings. Our trained automatic scorer shows a strong correlation (kappa=0.77) with human raters on the validation set. Meanwhile, according to the [Paper with Code](https://paperswithcode.com/sota/automated-essay-scoring-on-asap), the SOTA ASAP-AES model currently achieves kappa=0.791. Given that the average inter-human agreement is kappa=0.760 [1], we believe that our evaluation score is adequate to assess the quality of the generated text. Table 1 demonstrates that ChatGPT and GPT-4 generated essays receive higher quality ratings. Our empirical observations are that GPT-generated text generally surpasses the original STUDENT essays in terms of coherence, structure, and fluency. We plan to incorporate pairwise human evaluation in a future version (as per our reply to point 2 of Reviewer WQAB) and acknowledge the potential benefit of adding grammar checkers to further enhance quality.
> 2. Selection of topical words. As explained in Sec 3.2, LLMs have the tendency to excessively use topical words in the prompt. Therefore, replacing topical words could mislead the detectors by changing the word usage style of the generated essays. As explained in Sec 3.3.3 and "Algorithm 1 Word Substitun", we sort the key topic words based on their concurrency in the essay requirement (instruction) and replace them considering both LM and WordNet synonyms. Moreover, we provide experiments on the effect of word perturbation depth on detection accuracy, which is discussed in Sec 4.3.
> 3. Quality metric. As explained in Sec 4.1, our fine-tuned scorer achieves a kappa of 0.770, which closely aligns with the average inter-human agreement of kappa=0.760. Hence, it is deemed to provide a fair assessment of essay quality. We will consider more metrics, such as perplexity and grammar checker ratings for essay evaluation in a revision.
> 4. Dataset analysis. Thanks for the suggestion. As explained in lines 531-534, there are 8 topics with 3 different types. We should clarify that each LLM generates no less than 4,000 essays for each prompt with each perturbation method. Consequently, the collective result comprises a total of 110,535 generated essays with an average length of 247 words. We intend to incorporate a table of detailed dataset statistics.
> 5. Writing. Thanks for the suggestion. We will proofread and improve the writing.
>
> [1] Afrizal Doewes and Mykola Pechenizkiy. 2021. On the limitations of human-computer agreement in automated essay scoring. In EDM. International Educational Data Mining Society

---

### Official Review · Reviewer_4N4g · 2023-08-09

**Soundness:** 4

**Excitement:**

4: Strong: This paper deepens the understanding of some phenomenon or lowers the barriers to an existing research direction.

**Paper Topic And Main Contributions:**

The paper focuses on the task of detecting AI generated essays. The main contributions are as follows:
- dataset of AI generated (written, refined or continued by AI) essays, being a new benchmark for evaluating performance of AI generated text detectors in educational domain,
- attacks for detection of AI generated essays (paraphrase, sentence substitution, word substitution),
- broad evaluation of existing AIGC detectors on the created dataset.

**Questions For The Authors:**

Out of pure curiosity, is there any connection between the initial score of the essay from the ASAP dataset and the score after refinement or continuation by the model?

**Reasons To Accept:**

The paper is very well-written and presents a valuable contribution for detection of AIGC texts in educational domain. Moreover, the authors of the paper presented extensive evaluation of existing methods. They used both commercial and open-source models.

**Reasons To Reject:**

I can't think of any, it is a very good paper :)

**Reproducibility:**

5: Could easily reproduce the results.

**Reviewer Confidence:**

4: Quite sure. I tried to check the important points carefully. It's unlikely, though conceivable, that I missed something that should affect my ratings.

---

> ### Author Rebuttal · Authors · 2023-08-29
>
> Thanks for your constructive comments. Please see below for our responses.
>
> Question：
>
> As in reply to point 1 of Reviewer rqmL, it is essential to note that the original ASAP student essays were written by high-school students. Our empirical observations are that the GPT-generated text generally surpasses the original student essays regarding coherence, structure, and fluency. This is supported by the quality scores in Table 1, illustrating the refined essays consistently achieve higher quality scores compared to the continuation essays, which could be attributed to the fact that the continuation might have constrained the machine's creative writing style and direction.

---

### Official Review · Reviewer_9VU8 · 2023-08-11

**Soundness:** 4

**Excitement:**

4: Strong: This paper deepens the understanding of some phenomenon or lowers the barriers to an existing research direction.

**Missing References:**

Not that I know of.

**Paper Topic And Main Contributions:**

This paper studies several AIGC detectors’ performance on their new dataset, AIG-ASAP, which is specifically covering student essay writing. In the process of AIG-ASAP construction, authors employed 3 different automatic text perturbation techniques given human-written essay writings. Proposed text perturbation techniques range from coarse-grained modification (full-text level) to fine-grained modification levels (sentence/word-level), enabling the authors to investigate AIGC detectors’ vulnerabilities depending on perturbation depth. They also report that these perturbation depth affects AIGC detectors’ capabilities when they are finetuned on the curated dataset.

This paper’s contributions are as follows:
1. Curation of a new AIGC dataset (AIG-ASAP) in education domain
2. Introduction of three perturbation techniques that can effectively fool AIGC detectors
3. Providing the current state of state-of-the-art AIGC detectors on the curated dataset and highlighting their vulnerability

**Questions For The Authors:**

Q1. I’m curious why you used two different models (FLAN-T5 and BERT) for sentence and word substitution.
Q2. Have you tried evaluating the detectors’ performance in few shot manner? This could significantly influence models’ detection accuracy.
Q3. Sometimes LLMs are not faithful to the prompt. Did you put some efforts in checking the quality of generated texts? This may also impact the results (if generators were prone to be unfaithful to given prompts, the distinction between three different writing types will be blurred.)

**Reasons To Accept:**

Here are reasons to accept this paper:
1. This paper discusses a very timely and important topic given the surge in LLM usage in all writing domains.
2. AIGC detection in student essay writing is relatively under-explored compared to other writing domains.
3. It is well-structured and well-written, with few grammatical mistakes.
4. Its methodology and experiment setup with detector selection are reasonable.
5. It demonstrates several interesting findings that have not been introduced in other papers, such as the observations that detection difficulty varies to different essay types.

**Reasons To Reject:**

Here are weaknesses of this paper:
1. As stated in the paper, many existing papers have experimented with different text perturbation techniques. The existing works can serve baselines to gauge the effectiveness of proposed perturbation strategies more accurately.
2. The dataset seems limited.
3. For AIGC quality check, including humans’ evaluation sounds more reasonable and suitable than simply relying on the automated approach.

**Reproducibility:**

3: Could reproduce the results with some difficulty. The settings of parameters are underspecified or subjectively determined; the training/evaluation data are not widely available.

**Reviewer Confidence:**

4: Quite sure. I tried to check the important points carefully. It's unlikely, though conceivable, that I missed something that should affect my ratings.

**Typos Grammar Style And Presentation Improvements:**

Few grammatical mistakes I found:
In line 454, the reference text contains a parenthesis.
In line 492, both refined and paraphrasing perturbations requires -> require
ln line 513, ChatGPT paragraphing data -> paraphrasing data

---

> ### Author Rebuttal · Authors · 2023-08-29
>
> Thanks for your constructive comments. Please see below for our responses.
> 1. As discussed in Sec 3.3, the present research primarily centers around employing paraphrasers to counteract the detector's capabilities. For instance, in [1], content from paraphrasers is harmonized with human-generated sentences, while in [2], off-the-shelf models are harnessed for the rewriting process. Notably, these procedures share similarities to our "Essay Paraphrasing" methodology, leading us to believe that our perturbation method encompasses techniques from these preceding studies.
> 2. We agree that there is potential for expanding the dataset by incorporating additional LLMs or perturbation methods. Our current approach to data generation and perturbation was refined based on insights from multiple preliminary experiments. Nevertheless, it's worth noting that the AIG-ASAP dataset already stands at a scale ten times larger than the original ASAP dataset.  Also, compared to the corpus of ArguGPT, which consists of 4,038 argumentative generated essays, our dataset has a total of 110,535 generated essays.
> 3. Given the substantial volume of generated essays, we opted for an automated scoring model. However, as indicated in our response to Reviewer WQAB, we have a future plan to introduce a pairwise human evaluation approach.
>
> Q1: FLAN-T5 boasts greater potency and is relatively well-suited for generating lengthier sequences. Conversely, BERT, due to its compact size, exhibits limitations in text generation prowess. Employing BERT for word completion strikes a balance between effectiveness and efficiency.
>
> Q2: The detectors' performance in a few-shot manner has only marginal benefit over zero-shot detection in our preliminary experiments, so only the zero-shot and finetuned detection results are given. We are considering adding a few-shot evaluation in a future version.
>
> Q3: We did not systematically check the relevance of the generated essays to the prompts. However, our initial observation is that the LLMs are good at generating essays faithful to the prompts, and GPT-3.5/4 are in general better than Vicuna. We are considering adding more analysis. For instance, we will sample essays from the AIG-ASAP dataset to check, especially those that received low scores from the automatic scorer.
>
> [1] Kalpesh Krishna, Yixiao Song, Marzena Karpinska, John Wieting, and Mohit Iyyer. 2023. "Paraphrasing evades detectors of AI-generated text, but retrieval is an effective defense"
>
> [2] Vinu Sankar Sadasivan, Aounon Kumar, Sriram Balasubramanian, Wenxiao Wang, and Soheil Feizi. 2023. "Can ai-generated text be reliably detected?"

---

### Meta-Review · Area_Chair_JZhZ · 2023-09-17

**Recommendation:** 4

**Metareview:**

Reviewers agreed that the paper addresses an important new topic by creating a new dataset of perturbed student essays and highlighting the vulnerability of current AIGC detectors. They found the paper mainly to be sound and well-executed, although there were suggestions that including a human evaluation would be helpful and more baselines from previous work could be included. Other criticism concerned improvements in the presentation of the content that could easily be fixed in a final version.

---

### Decision · Program_Chairs · 2023-10-07

**Decision:**

Accept-Main

**Comment:**

Reviewers agreed that the paper addresses an important new topic by creating a new dataset of perturbed student essays and highlighting the vulnerability of current AIGC detectors. They found the paper mainly to be sound and well-executed, although there were suggestions that including a human evaluation would be helpful and more baselines from previous work could be included. Other criticism concerned improvements in the presentation of the content that could easily be fixed in a final version.